# A Systematic Review and Meta-Analysis of the Implementation of High-Performance Cardiopulmonary Resuscitation on Out-of-Hospital Cardiac Arrest Outcomes

**DOI:** 10.3390/jcm10102098

**Published:** 2021-05-13

**Authors:** Qin Xiang Ng, Ming Xuan Han, Yu Liang Lim, Shalini Arulanandam

**Affiliations:** Emergency Medical Services Department, Singapore Civil Defence Force, 91 Ubi Ave 4, Singapore 408827, Singapore; mxhan9598@yahoo.com (M.X.H.); yulianglim95@gmail.com (Y.L.L.); Shalini_ARULANANDAM@scdf.gov.sg (S.A.)

**Keywords:** cardiopulmonary resuscitation, CPR, prehospital care, resuscitation, emergency medical services, EMS, paramedicine

## Abstract

Despite numerous technological and medical advances, out-of-hospital cardiac arrests (OHCAs) still suffer from suboptimal survival rates and poor subsequent neurological and functional outcomes amongst survivors. Multiple studies have investigated the implementation of high-quality prehospital resuscitative efforts, and across these studies, different terms describing high-quality resuscitative efforts have been used, such as high-performance CPR (HP CPR), multi-tiered response (MTR) and minimally interrupted cardiac resuscitation (MICR). There is no universal definition for HP CPR, and dissimilar designs have been employed. This systematic review thus aimed to review current evidence on HP CPR implementation and examine the factors that may influence OHCA outcomes. Eight studies were systematically reviewed, and seven were included in the final meta-analysis. Random-effects meta-analysis found a significantly improved likelihood of prehospital return of spontaneous circulation (pooled odds ratio (OR) = 1.46, 95% CI: 1.16 to 1.82, *p* < 0.001), survival-to-discharge (pooled OR = 1.32, 95% CI: 1.16 to 1.50, *p* < 0.001) and favourable neurological outcomes (pooled OR = 1.24, 95% CI: 1.11 to 1.39, *p* < 0.001) with HP CPR or similar interventions. However, the studies had generally high heterogeneity (I^2^ greater than 50%) and overall moderate-to-severe risk for bias. Moving forward, a randomised, controlled trial is necessary to shed light on the subject.

## 1. Introduction

Of the myriad prehospital challenges confronting emergency medical services (EMS) systems worldwide, out-of-hospital cardiac arrests (OHCAs) remain a global challenge [1,2,3,4,5]. Despite systemic improvements in training, and the upgrading of equipment and skill sets, survival rates remain disappointingly low [1,2,3]. A recent study estimated the global OHCA survival rate to be an average of 5% to 10%, with only slight improvements over the years and also poor subsequent neurological and functional outcomes amongst survivors [4].

Interruptions to chest compressions persistently emerge as a major contributor to the high mortality associated with OHCAs [6]. The efficacy of prehospital interventions is majorly influenced by time sensitivity, as highlighted in the chain of survival. This chain spells out the sequence of events that strictly inform the management of OHCAs, namely early access, early cardiopulmonary resuscitation (CPR), early defibrillation, early advanced life support and early post-resuscitative care [7,8]. There is a strong emphasis on the timely detection, swift community response and subsequent execution of life support measures in order to augment survival rates [9].

The ongoing coronavirus disease 2019 (COVID-19) pandemic has significantly impeded each link in the chain of survival [10,11], providing an impetus for further OHCA research and innovation. In the age of COVID-19, global EMS systems have reported increases in the incidence of OHCAs and worse outcomes. A systematic review of 10 studies across five countries reported a “more than 2-fold” increase in the OHCA incidence during the pandemic [7]. Similarly, a Singaporean study also reported an increase in the OHCA incidence and inferior outcomes, results that were in line with those of EMS systems in Europe, New York City and Victoria [8]. These findings in the current pandemic era further underscore the pressing need to evaluate current and newer strategies to strengthen OHCA prehospital systems of care.

In recent years, there is growing interest in the advantages and implementation of high-quality prehospital OHCA resuscitative efforts. Multiple approaches as to what defines high quality have been used. This includes high-performance CPR (HP CPR), multi-tiered response (MTR) and minimally interrupted cardiac resuscitation (MICR) [12,13,14,15,16]. The approaches elaborated in these studies are united in their common goal to provide quality resuscitative efforts by focusing on responder skills and intra-professional role coordination. 

Yet, these studies together have not managed to provide a conclusive answer on the optimal EMS team configuration or ascertain that high-quality resuscitative efforts, henceforth labelled as HP CPR, can provide the holy grail to better outcomes for OHCAs. The implementation of HP CPR in some EMS systems has found nonsignificant differences for prehospital return of spontaneous circulation (ROSC) in some instances [16]. There is also no universal definition for what constitutes HP CPR, though it is generally agreed that HP CPR involves a coordinated team-based resuscitation, where each team member is assigned a specific task. It is therefore imperative to review the current evidence on HP CPR implementation to substantiate its impact on OHCA outcomes and influence further efforts by EMS systems to make it a standard of care. 

## 2. Methods

A systematic literature search was performed in accordance with the latest Preferred Reporting Items for Systematic Reviews and Meta-Analyses (PRISMA) guidelines [17]. Using the keywords “high-performance CPR OR HP CPR OR HPCPR OR HP-CPR OR team CPR,” a preliminary search on the PubMed, OVID Medline, Embase, ScienceDirect, Clinicaltrials.gov, Web of Science, and Google Scholar databases yielded 208 papers published in English between 1 January 1988 and 31 March 2021. To maximise sensitivity, the search strategies relied on blended subject headings and keyword (free text) approaches. Attempts were made to search the gray literature as well using the Google search and hand searching. Title/abstract screening was performed independently by two researchers (Q.X.N. and M.X.H.) to identify articles of interest. For relevant abstracts, full articles were obtained, reviewed and also checked for references of interest. If necessary, the authors of the articles were contacted to provide additional data.

Full articles were obtained for all selected abstracts and reviewed by three researchers (Q.X.N., M.X.H. and Y.L.L.) for inclusion. Any disagreement was resolved by discussion and consensus. The inclusion criteria for this review were as follows: (i) original published study, (ii) implementation of HP CPR or similar minimally interrupted CPR or a multi-tiered response system and (iii) reporting of OHCA outcomes. Non-human studies, abstracts and conference proceedings were excluded from review. The primary outcome measures of interest were neurological outcomes, survival-to-discharge and ROSC rates in patients who received an HP CPR intervention or similar interventions as opposed to controls. Data such as study design, sample size and study population were extracted from the studies reviewed and are summarised in Table 1. Odds ratios (ORs) comparing the neurological recovery, survival-to-discharge and ROSC rates in the intervention (HP CPR) and control (or historical comparator in some cases) groups were calculated. Estimates were pooled, and where appropriate, 95% confidence intervals (95% CIs) and *p*-values were calculated. Heterogeneity among the different studies pooled was examined using the I^2^ statistic and Cochran’s Q test. Publication bias was assessed using a funnel plot and the Egger test [18]. All analyses were conducted using MedCalc statistical software version 14.8.1 (MedCalc Software bvba, Ostend, Belgium).

The quality and risk of bias of the studies were also assessed with the Risk Of Bias In Non-randomized Studies—of Interventions (ROBINS-I) tool [19], graded based on the consensus of three study investigators (Q.X.N., M.X.H. and Y.L.L.).

## 3. Results

As seen in Figure 1, a total of eight studies were included in this systematic review ref. [12,13,14,15,16,20,21,22]. Most of these were retrospective cohort studies. One study [13] was excluded from the final meta-analysis as it lacked a control group for comparison. Unlike the other studies, in the study by Fang et al., the authors calculated ORs based on a comparison of different crew numbers and the EMT–paramedic ratio was treated as a continuous variable: 25.0–33.3%, 50%, 66.7–75.0% and 100% [13].

Using a random-effects model (as I^2^ is greater than 50%), the forest plot (Figure 2) showed that the pooled OR for survival-to-discharge with HP CPR or similar intervention was 1.32 (95% CI: 1.16 to 1.50, *p* < 0.001), supporting a significantly improved likelihood of survival.

The benefits of HP CPR seem to lie in maximising hands-on-chest time, early defibrillation, early advanced life support, early advanced airway management and early administration of resuscitative drugs.

When examining the pooled OR for any ROSC (Figure 3), there was also an increased likelihood of any ROSC with HP CPR or similar intervention (pooled OR 1.46, 95% CI: 1.16 to 1.82, *p* < 0.001).

Only three studies examined neurological outcomes (graded using the CPC) with HP CPR or similar intervention. The pooled OR of 1.24 (95% CI: 1.10 to 1.39, *p* < 0.001) also supports significantly better neurological recovery with such interventions (Figure 4).

Visual examination of the funnel (Figure 5) and Egger test did not indicate the presence of funnel plot asymmetry (*p* = 0.0065); however, the reliability of the Egger test was limited by the small number of studies, that is, less than 10 studies [18]. No sensitivity analysis was performed due to the small number of available studies.

As for the risk of bias of the various studies (as shown in Table 2), most had overall moderate-to-serious risk due to the lack of control for unmeasured confounders, such as geographical factors, community characteristics and patient-related factors, e.g., baseline health status.

## 4. Discussion

Overall, the studies support the implementation of high-quality resuscitative efforts to improve OHCA outcomes in terms of prehospital ROSC attainments, survival-to-discharge and neurological recovery of survivors. However, the studies had differing operating contexts and EMS configurations. Most were conducted in metropolitan rather than rural settings and had a paramedic crew size of at least four EMTs. As EMS organisations are constrained by manpower and other finite resources, it is important to review the HP CPR team composition.

There is burgeoning research on HP CPR, which is delivered via team-based resuscitation where each team member is assigned a specific task. In a North American study [22], the positive benefits from the implementation of an MTR was more significant in OHCA patients with an initial shockable rhythm; hence, the authors were of the opinion that early defibrillation enabled by an MTR is vital to improve outcomes. 

In a South Korean study of MTRs [20], an early MTR (defined as 0–18 min from call to second EMS arrival) was essential to improve neurological outcomes and survival-to-discharge compared to the single-tiered response group or late MTR group (19 min or longer from call to second EMS arrival). In fact, a late MTR had slightly better prehospital ROSC rates than the single-tiered response system but worse survival and neurological outcomes. This is perhaps expected, given that previous studies have confirmed the time-sensitive nature of OHCAs [23,24]. Advanced life support (ALS) arrival within 10 min from call (the optimal threshold) is associated with improved outcomes in OHCA patients [24], and having more EMS responders arrive within 15 min of the call is also associated with higher survival [22]. 

In addition to numbers, the skill level of the EMS team is also an important consideration. As highlighted by studies from Taiwan [13,21], the EMT–paramedic ratio but not the number of EMTs is positively associated with survival [21]. This observation was corroborated by a more recent Taiwanese study [13]; the training hours required for EMT paramedic accreditation in Taiwan is 1280 h, which is more than four times that of the EMT intermediate accreditation. Moreover, the EMT paramedic is also able to perform more advanced procedures, e.g., endotracheal intubation, manual defibrillation, intravascular therapy and transcutaneous pacing, which the EMT intermediate cannot. In an analysis of the national MTR rollout in Korea over 2 years [14], the authors found significantly improved prehospital ROSC outcomes as the MTR matured over time, and attributed this to the increased provision of IV drugs and advanced airway management by the paramedics. Time-to-first defibrillation and time-to-first intravenous (IV) epinephrine are clearly key interventions that should be prioritised, and team members must have the skills to perform these.

There is yet another aspect of HP CPR execution that is worthy of deliberation. Co-interventions such as mechanical CPR, traditionally regarded as a key component of the resuscitation algorithm, might not be so significant after all. A recent Victorian study specified that the local EMS system had discouraged the use of mechanical CPR during the crucial early stages of resuscitation [16]. This reduced use of mechanical CPR decreased interruptions to chest compressions during HP CPR. This de-emphasis on mechanical CPR is supported by a recent systematic review of controlled and uncontrolled trials that revealed no meaningful change in survival outcomes when mechanical CPR was applied [25]. Another Australian study even revealed increased incidence of airway haemorrhage in non-traumatic OHCA cases, which would adversely impact survival [26]. This suggests that the Victorian method of focusing on minimal interruptions by maximising the hands-on-chest time without mechanical CPR could inform better HP CPR configurations.

The present study has several strengths. Firstly, to date, this is the first systematic review and meta-analysis that has attempted to investigate the impact of HP CPR and related interventions on ROSC and survival rates. By extension, our results could provide a platform for comparisons across EMS systems internationally and generate further discussions on the subject. In addition, this study has shown that (1) HP CPR can lead to improved ROSC rates and survival-to-discharge and (2) the HP CPR team composition matters. This could drive changes in paramedic training and protocol development in order to improve patient outcomes. 

The potential limitations of this study have to be acknowledged. Firstly, the review protocol was not prospectively registered. Secondly, given the different operating contexts and team configurations of the EMS systems covered in this study, we are still admittedly unclear on what the best configuration is for HP CPR. This would depend on the local context and geography. Thirdly, there was significant heterogeneity observed in the meta-analysis, as seen in the forest plots and reflected by the generally high I^2^ statistic (>50%). This could be attributed to differences in the various EMS configurations, the population under study, the providers’ training standards and corresponding CPR quality. The research designs across the included studies were also not homogenous as different EMS systems have a different number of team members and composition. Fourthly, only three studies reflected the secondary outcome of neurological status. Finally, as most of the included studies employed an observational design, it was difficult to adjust for potential confounding factors such as event-related factors and patient demographics. The studies had an overall moderate-to-serious risk of bias, and the findings must be interpreted in light of these shortcomings. 

## 5. Conclusions

Current evidence suggests that HP CPR, MTR and similar interventions significantly improve prehospital ROSC, survival-to-discharge and neurological recovery for OHCA patients. However, the operating context matters, and the optimal EMS team configuration remains unknown. The skill level and EMT-to-paramedic ratio plays a part as well. More paramedics rather than more EMTs, with their additional armamentarium of drugs and skills, may make more difference than the actual increased hands-on-chest time afforded by non-EMS-heavy teams. Realistically, HP CPR and the MTR may be more resource-intensive than a single-tiered response system and may be difficult to support as EMS utilisation continues to increase. It is thus important to accurately triage and identify OHCA cases early. Future studies should also measure and control for community, patient and hospital characteristics. A randomised, controlled trial design might help shed more light on the topic as well.

## Figures and Tables

**Figure 1 jcm-10-02098-f001:**
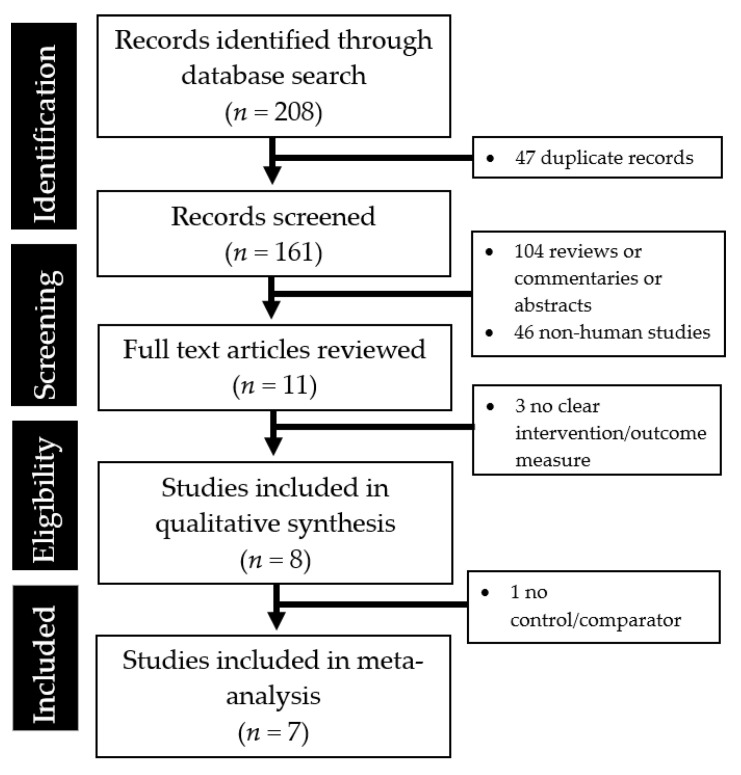
PRISMA flow diagram showing the studies identified during the literature search and abstraction process.

**Figure 2 jcm-10-02098-f002:**
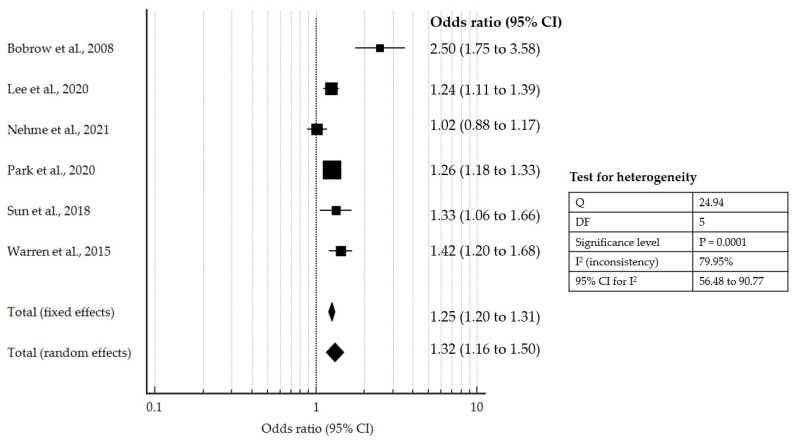
Forest plot showing pooled OR for survival-to-discharge with HP CPR or similar intervention compared to controls [12,14,16,20,21,22].

**Figure 3 jcm-10-02098-f003:**
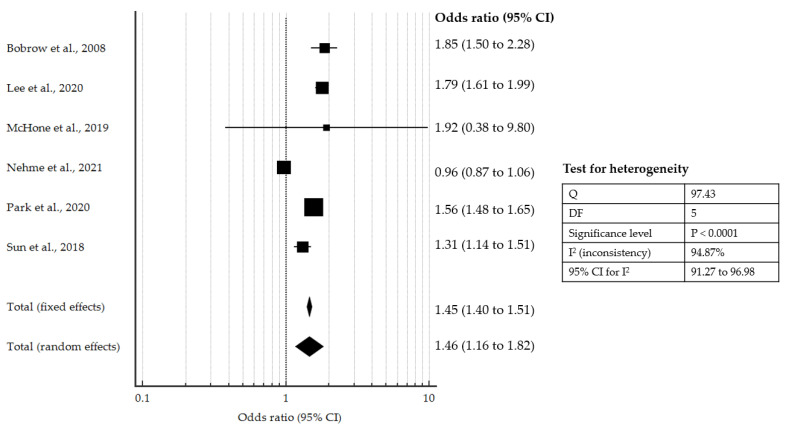
Forest plot showing pooled OR for any ROSC with HP CPR or similar intervention compared to controls [12,14,15,16,20,21].

**Figure 4 jcm-10-02098-f004:**
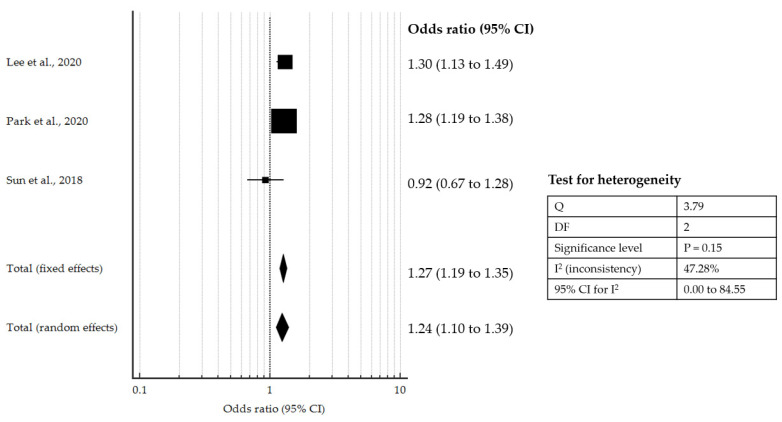
Forest plot showing pooled OR for good neurological outcomes with HP CPR or similar intervention compared to controls [14,20,21].

**Figure 5 jcm-10-02098-f005:**
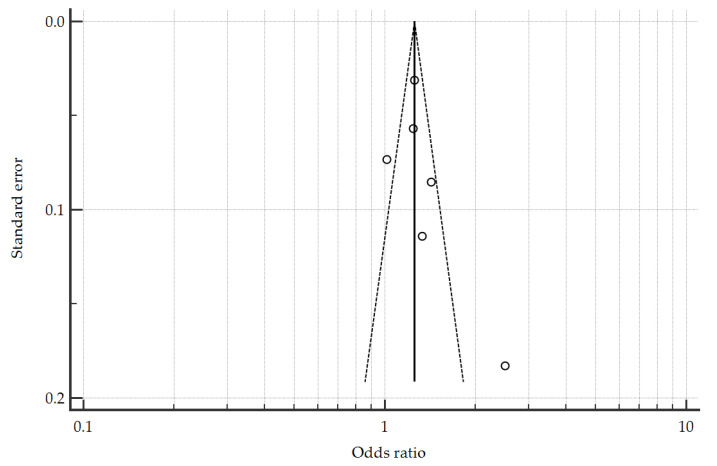
Funnel plot (with pseudo 95% confidence intervals) to assess publication bias; Egger test for publication bias = 2.047, 95% CI: 0.955 to 3.14, *p* = 0.0065.

**Table 1 jcm-10-02098-t001:** Available studies on the implementation of HP CPR or similar interventions and OHCA outcomes.

Study, Year	Country	Study Design and Sample Size (*N*)	Intervention Type and Controls for Comparison	Outcome Measures	Odds Ratios	Conclusions
Bobrow et al., 2008 [12]	United States	Prospective review of OHCAs in two metropolitan cities in Arizona (*N* = 886)	Protocol type: before MICR training versus after MICR training	Primary outcomes: survival-to-hospital discharge, survival with witnessed VF Secondary outcomes: ROSC, survival-to-hospital admission	Survival-to-hospital discharge: aOR 3.0 (95% CI: 1.1 to 8.9) Survival with witnessed VF: aOR 8.6 (95% CI: 1.8 to 42.0) ROSC: aOR 1.3 (95% CI: 0.8 to 2.0) Survival-to-hospital admission: aOR 0.8 (95% CI: 0.5 to 1.2)	Overall, survival-to-hospital discharge increased from 1.8% before MICR training to 5.4% after MICR training; the greatest improvement was seen for cases with documented witnessed cardiac arrest and a shockable initial arrest rhythm.
Fang et al., 2020 [13]	Taiwan	Retrospective cohort (*N* = 1357)	Skill level: higher EMT–paramedic ratio versus lower EMT–paramedic ratio	Primary outcome: sustained (>2 h) ROSC Secondary outcomes: any ROSC, survival-at-hospital-discharge, favourable neurologic status (CPC level I and II at discharge)	Sustained ROSC: aOR 1.08 (95% CI: 1.02 to 1.13) Survival-to-discharge: aOR 1.23 (95% CI: 0.82 to 1.84) Favourable neurological outcome at discharge: aOR 1.12 (95% CI: 1.01 to 1.26)	An increased EMT–paramedic ratio but not number of on-scene EMTs was linked to improved ROSC and neurological outcomes.
Lee et al., 2020 [14]	South Korea	Naturalistic cohort (*N* = 32,663)	Crew numbers: more on-scene EMS providers versus on-scene fewer EMS providers; classified as *A-MTR* if an additional ambulance was dispatched or *F-MTR* if an additional fire engine was dispatched	Primary outcome: prehospital defibrillation of OHCA patients Secondary outcomes: prehospital ROSC, survival-to-discharge, good neurological outcome (CPC level I and II at discharge)	Prehospital defibrillation: aOR 1.16 (95% CI: 1.08 to 1.25) Prehospital ROSC: aOR 1.82 (95% CI: 1.63 to 2.04) Survival-to-discharge: aOR 1.37 (95% CI: 1.21 to 1.56) Good neurological outcome: aOR 1.23 (95% CI: 1.06 to 1.43)	Over a 2-year study period, as the multi-tiered response (MTR) intervention matured, the rate of prehospital defibrillation, prehospital ROSC, survival-to-discharge and good neurological outcomes also improved. The MTR group also provided more advanced airway and intravenous drug management.
McHone et al., 2019 [15]	United States	Pre- and post-implementation retrospective cohort (*N* = 24)	Protocol type: before TF-HP-CPR (an approach that emphasises early defibrillation, ample duty-rest cycles and BVM or BIAD use) protocol implementation versus after TF-HP-CPR protocol implementation	Primary outcome: prehospital ROSC Secondary outcome: documentation of end-tidal carbon dioxide values	Prehospital ROSC: OR 1.92 (95% CI: 0.376 to 9.80)	The implementation of a team-focused HP CPR protocol in a rural-area EMS improved the rate of prehospital ROSC among patients with OHCA, albeit not statistically significant (*p* = 0.682).
Nehme et al., 2021 [16]	Australia	Interrupted time-series analysis (*N* = 10,600)	Protocol type: intervention period (HP CPR resuscitation, mCPR discouraged) versus control period (ARC guidelines)	Primary outcome: survival-to-hospital discharge Secondary outcomes: event survival, prehospital ROSC	Survival-to-hospital discharge: aOR 1.33 (95% CI: 1.11 to 1.58) Event survival: aOR 1.34 (95% CI: 1.09 to 1.65) Prehospital ROSC: aOR 1.38 (95% CI: 1.14 to 1.65)	After a 12-month intervention period, the implementation of an HP CPR programme improved OHCA survival.
Park et al., 2020 [20]	South Korea	Prospective cross-sectional study (*N* = 54,436)	Crew numbers: more on-scene EMS providers fewer on-scene EMS providers Single-tiered: ambulance only Early MTR: ambulance and fire engine or 2 ambulances, which responded within 18 min. Late MTR: ambulance and fire engine or 2 ambulances, that responded after 18 min.	Primary outcome: good neurological outcome (CPC level I and II at discharge) Secondary outcomes: survival-to-hospital discharge, prehospital ROSC	Good neurological outcome: aOR 1.15 (95% CI: 1.06 to 1.26) Survival-to-discharge: aOR 1.13 (95% CI: 1.06 to 1.21) Prehospital ROSC: aOR 1.46 (95% CI: 1.38 to 1.56)	Early MTR improved neurological outcomes and survival-to-discharge compared to the single-tiered response group or late MTR.
Sun et al., 2018 [21]	Taiwan	Retrospective cohort (*N* = 8262)	Skill level: higher EMT–paramedic ratio versus lower EMT-paramedic ratio	Primary outcome: survival-to-hospital discharge Secondary outcome: good neurological outcome at discharge (CPC level I and II)	Survival-to-discharge: aOR 1.36 (95% CI: 1.06 to 1.76) Sustained ROSC: aOR 1.17 (95% CI: 1.00 to 1.37) Good neurological outcome: aOR 1.26 (95% CI: 0.86 to 1.83)	An increased on-scene EMT–paramedic ratio >50% significantly improved survival-to-discharge and neurological outcomes for OHCA cases, especially for those with witnessed, non-shockable rhythm.
Warren et al., 2015 [22]	Canada and United States	Retrospective cohort (*N* = 16,122)	Crew numbers: more on-scene EMS personnel versus fewer on-scene EMS personnel	Primary outcome: survival-to-discharge	Survival-to-discharge: aOR 1.35 (95% CI: 1.05 to 1.73)	Compared to the reference number of 5 or 6 on-scene EMS personnel, 7 or 8 on-scene EMS personnel, within 15 min of call, were associated with significantly improved survival. The benefits were unlikely solely due to early CPR or defibrillation.

Abbreviations: aOR, adjusted odds ratio; ARC, Australian Resuscitation Council; BIAD, blind insertion airway device; BVM, bag valve mask; CI, confidence interval; CPC, cerebral performance category; CPR, cardiopulmonary resuscitation; EMT, emergency medical technician; EMS, emergency medical services; HP CPR, high-performance CPR; mCPR, mechanical CPR; MICR, minimally interrupted cardiac resuscitation; MTR, multi-tiered response; OHCA, out-of-hospital cardiac arrest; ROSC, return of spontaneous circulation; VF, ventricular fibrillation.

**Table 2 jcm-10-02098-t002:** Risk of bias assessment with the ROBINS-I tool.

Study	Confounding	Selection	Measurement of Intervention	Missing Data	Measurement of Outcomes	Reported Result	Overall
Bobrow et al., 2008 [12]	Moderate	Moderate	Low	Low	Low	Low	Moderate
Fang et al., 2020 [13]	Serious	Serious	Moderate	Moderate	Moderate	Low	Serious
Lee et al., 2020 [14]	Serious	Serious	Low	Moderate	Low	Moderate	Moderate
McHone et al., 2019 [15]	Serious	Critical	Moderate	Serious	Serious	Serious	Serious
Nehme et al., 2021 [16]	Serious	Moderate	Low	Moderate	Low	Moderate	Moderate
Park et al., 2020 [20]	Moderate	Moderate	Low	Low	Low	Moderate	Moderate
Sun et al., 2018 [21]	Moderate	Moderate	Moderate	Moderate	Low	Moderate	Moderate
Warren et al., 2015 [22]	Serious	Moderate	Moderate	Serious	Moderate	Moderate	Serious

## Data Availability

The datasets generated and/or analysed during the current study are available from the corresponding author on reasonable request.

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
