# Peer review of "A Systematic Review and Meta-Analysis of the Implementation of High-Performance Cardiopulmonary Resuscitation on Out-of-Hospital Cardiac Arrest Outcomes"

_jcm, 2021, doi:10.3390/jcm10102098_

Round 1

Reviewer 1 Report

Here, Dr. Ng and colleagues performed a systematic review of 8 studies to better understand the impact of high performance (HP) CPR on the outcome of OHCA. As the results, not-so-surprisingly HP CPR improved OHCA outcomes. In general, the systematic review was conducted properly and the manuscript is well-written. However, several comments and concerns need to be addressed by the authors:

  • The main "limitation" of this manuscript is that its protocol was not registered in PROSPERO (https://www.crd.york.ac.uk/prospero/).
  • The PRISMA checklist was not included in the submission.
  • Please be clear whether this is "only" a systematic review as stated in the title, or this is a systematic review and meta-analysis as shown in the PRISMA flowchart. Please adapt the title.
  • I don't see the clear definition or criteria for HP CPR in the introduction? Please define it clearly because these criteria are important for selecting appropriate studies.
  • In some studies included in the meta-analysis (REF #21 and #22), they studied the number of emergency medical personnelles on the outcome of OHCA. Is this also a criterion of HP CPR? Please clarify whether these studies are appropriate.
  • Please avoid using exaggerated words, such as "dismal", "plague", "meagre", etc. Please replace those words with more common scientific words. In general, the introduction needs to be better arranged and the writing style needs to be improved following the common guide for scientific writing.
  • This kind of sentence "However, any chain is only as strong as its weakest link" is acceptable for an editorial or a perspective article but not for this type of article (e.g., original research or reviews).
  • I am not sure if I understand correctly about the inclusion and exclusion. The authors in the flowchart said that they removed 1 study from the systematic review because no control / comparator but in the table I can see the OR values in all 8 included studies. Please clarify which study was excluded and describe it adequately in the manuscript.
  • In lines 13-14, the authors explained that the motivation for this systematic review is because the result of the studies investigating HP CPR and OHCA yielded conflicting conclusions. But this was not reflected in the forest plots. They seem to be consistent (no study has OR below 1) as also stated by the authors in lines 151-152. Please justify the arguments.
  • Line 9: remove the hyphen from "sub-sequent"
  • Please replace "chinese taipei" with "taiwan" if the authors agree that they are interchangeable.
  • Please include the OR and the 95% CI of each study to the forest plots. The authors can benefit from the previous example (https://adc.bmj.com/content/90/8/845)

Author Response

Comment 1: The main "limitation" of this manuscript is that its protocol was not registered in PROSPERO (https://www.crd.york.ac.uk/prospero/).

  • Thank you for the comment. We have acknowledged this in our discussion of study limitations.

Comment 2: The PRISMA checklist was not included in the submission.

  • Thank you for the comment. We apologise for the oversight and have now appended the PRISMA checklist under supplementary material.

Comment 3: Please be clear whether this is "only" a systematic review as stated in the title, or this is a systematic review and meta-analysis as shown in the PRISMA flowchart. Please adapt the title.

  • Thank you for the comment. We adapted the title of our submission as advised.

Comment 4: I don't see the clear definition or criteria for HP CPR in the introduction? Please define it clearly because these criteria are important for selecting appropriate studies.

  • Thank you for the comment. Unfortunately, there is also no universal definition for what constitutes HP CPR, though it is generally agreed that HP CPR involves a coordinated team-based resuscitation, where each team member is assigned a specific task. We have now highlighted this in the introduction.

Comment 5: In some studies included in the meta-analysis (REF #21 and #22), they studied the number of emergency medical personnel on the outcome of OHCA. Is this also a criterion of HP CPR? Please clarify whether these studies are appropriate.

  • Thank you for the comment. As aforementioned, there is also no universal definition for what constitutes HP CPR, though it is generally agreed that HP CPR involves a coordinated team-based resuscitation, where each team member is assigned a specific task. Thus, these studies were still considered appropriate for inclusion in the present review.

Comment 6: Please avoid using exaggerated words, such as "dismal", "plague", "meagre", etc. Please replace those words with more common scientific words. In general, the introduction needs to be better arranged and the writing style needs to be improved following the common guide for scientific writing.

  • Thank you for the comments. We have now done a close edit of the entire manuscript for language and style.

Comment 7: This kind of sentence "However, any chain is only as strong as its weakest link" is acceptable for an editorial or a perspective article but not for this type of article (e.g., original research or reviews).

  • Thank you for the comment. We have omitted the superfluous description as suggested.

Comment 8: I am not sure if I understand correctly about the inclusion and exclusion. The authors in the flowchart said that they removed 1 study from the systematic review because no control / comparator but in the table I can see the OR values in all 8 included studies. Please clarify which study was excluded and describe it adequately in the manuscript.

  • Thank you for the comment. We apologise for the confusion. We have now clarified that, “Unlike the other studies, in the study by Fang et al., the authors calculated ORs based on a comparison of different crew numbers and EMT-paramedic ratio was treated as a continuous variable, 25.0–33.3%, 50%, 66.7–75.0%, and 100% [13].” Hence it was excluded from the meta-analysis.

Comment 9: In lines 13-14, the authors explained that the motivation for this systematic review is because the result of the studies investigating HP CPR and OHCA yielded conflicting conclusions. But this was not reflected in the forest plots. They seem to be consistent (no study has OR below 1) as also stated by the authors in lines 151-152. Please justify the arguments.

  • Thank you for the comment. We agree with the reviewer and have modified our arguments to read, “The implementation of HP CPR in some EMS systems have found nonsignificant differences for prehospital return of spontaneous circulation (ROSC) in some instances [16].”

Comment 10: Line 9: remove the hyphen from "sub-sequent"

  • Thank you for the comment. We have corrected this typo.

Comment 11: Please replace "chinese taipei" with "taiwan" if the authors agree that they are interchangeable.

  • Thank you for the comment. We replaced all mentions of “Chinese Taipei” with “Taiwan” as advised.

Comment 12: Please include the OR and the 95% CI of each study to the forest plots. The authors can benefit from the previous example (https://adc.bmj.com/content/90/8/845)

  • Thank you for the comment. We have now included the OR and corresponding 95% CI for all forest plots.

Reviewer 2 Report

Qin Yiang Ng, et al.: A Sytematic Review of the Implementation of High-Performance Cardiopulmonary Resuscitation on Out-of-Hospital Cardiac Arrest Outcomes

Qin Xiang Ng et al. performed a systematic review about the outcomes of out-of-hospital cardiac arrest (OHCAs) after the implementation of high-performance cardiopulmonary resuscitation. In a systematic literature search, a total of 208 abstracts were found, which were screened by two independent researchers to identify those for inclusion in the systematic review, resulting in the inclusion of eight full text articles in the present review, of which one was excluded afterwards due to the lack of a control for comparison. Primary outcome measures were the rate of return of spontaneous circulation, survival-to-discharge and the neurological outcomes of patients with OHAOs. In Table 1 the studies included in the present analysis with study design, number of patients included, outcome measures, OR and conclusion are listed. In Figure 2 to 4 forest plots showing the OR for the primary outcomes are shown. High-performance CPR improved OHCA outcomes in term of return of spontaneous circulation, survival-to-discharge (both with 6 studies included), and neurological outcomes (3 studies included). However, a high heterogeneity was found, with I2 ranging from around 50 to 95%.

I have some points to consider:

  1. Abstract: By structuring the abstract it would be more easy to catch the results and the conclusion of the manuscript.
  2. Results: A review should contain all relevant information so that it enables the reader to get a clear overview of the topic and the results of the manuscript. In Table 1, key points about the studies included in the present analysis are given. However, some information about the HP-CPR in each of the studies (differences in different countries) and about the control group would be desirable.
  3. Figures: Please add OR (95% CI) to the figures for each study.
  4. Discussion: There are several points addressed in the discussion section which are interesting for the reader and might be presented in the result section to give a clear picture about benefits/options and, in addition, for discouraging results of HP-CRP (i.e. mechanical CPR).
  5. Conclusion: The authors state that “the benefits of HP CPR seem to lie in maximizing hands-on-chest time, early defibrillation, early advanced life support, early advanced airway management  and early administration of resuscitative drugs.” To this point, these benefits have not been shown in the result section and thus, should not be concluded from the results reported. However, these points might be conclusions from the single studies/HP CPR, and if so, should be demonstrated/itemized in the result section. 
  6. The key message of the review is not quite clear. What is the eminent information about the OHCAs and the benefit for the paramedics and thus the patient. 

Author Response

Comment 1: Abstract: By structuring the abstract it would be more easy to catch the results and the conclusion of the manuscript.

  • Thank you for the comment. We have better structured our abstract, however, as per the journal’s guidelines, no headings were included. “The abstract should be a single paragraph and should follow the style of structured abstracts, but without headings.”

Comment 2: Results: A review should contain all relevant information so that it enables the reader to get a clear overview of the topic and the results of the manuscript. In Table 1, key points about the studies included in the present analysis are given. However, some information about the HP-CPR in each of the studies (differences in different countries) and about the control group would be desirable.

  • Thank you for the comment. We have now added more information about the intervention and control group in Table 1.

Comment 3: Figures: Please add OR (95% CI) to the figures for each study.

  • Thank you for the comment. We have now included the OR and corresponding 95% CI for all forest plots.

Comment 4: Discussion: There are several points addressed in the discussion section which are interesting for the reader and might be presented in the result section to give a clear picture about benefits/options and, in addition, for discouraging results of HP-CRP (i.e. mechanical CPR).

  • Thank you for the comment. We have modified our results section as suggested.

Comment 5: Conclusion: The authors state that “the benefits of HP CPR seem to lie in maximizing hands-on-chest time, early defibrillation, early advanced life support, early advanced airway management and early administration of resuscitative drugs.” To this point, these benefits have not been shown in the result section and thus, should not be concluded from the results reported. However, these points might be conclusions from the single studies/HP CPR, and if so, should be demonstrated/itemized in the result section.

  • Thank you for the comment. We have tempered our conclusions as suggested.

Comment 6: The key message of the review is not quite clear. What is the eminent information about the OHCAs and the benefit for the paramedics and thus the patient?

  • Thank you for the comment. We believe that our review has two main takeaways. First, the review showed that HP CPR and related interventions can lead to improved ROSC rates and survival-to-discharge, and second, the HP CPR team composition matters. This could drive changes in paramedic training and protocol development, in order to improve patient outcomes. More paramedics rather than more EMTs, with their additional armamentarium of drugs and skills, may make more difference than the actual increased hands-on-chest time afforded by non EMS-heavy teams.

Round 2

Reviewer 1 Report

Thanks for addressing my comments, which has significantly improved the manuscript.